# Socioeconomic gradient of lean diabetes in India: Evidence from National Family Health Survey, 2019–21

**Surama Manjari Behera**[1‡], **Priyamadhaba Behera**[2‡], **Sanjay K. Mohanty**[3], **Rajeev Ranjan Singh**[4*], **Binod Kumar Patro**[5], **Avinaba Mukherjee**[6], **Venkatarao Epari**[7]

**1** Siksha 'O' Anusandhan (SOA University), Bhubaneswar, India, **2** Dept of Community Medicine & Family Medicine, All India Institute of Medical Sciences, Bhubaneswar, India, **3** Department of Population and Development, International Institute for Population Sciences, Mumbai, India, **4** International Institute for Population Sciences, Mumbai, India, **5** Professor of Community Medicine & Dean, Dept of Community Medicine & Family Medicine, All India Institute of Medical Sciences, Bhubaneswar, India, **6** Dept of Community Medicine & Family Medicine, All India Institute of Medical Sciences, Bhubaneswar, India, **7** Department of Community Medicine, Siksha 'O' Anusandhan (SOA University), Bhubaneswar, India

‡ SMB and PB are joint first author on this work.
* rajeevs210@gmail.com

**Data Availability Statement:** Data for this study were extracted from the fifth round of the National family health survey of India 2019-2021, which is

## Abstract

Diabetes is a global public health challenge, particularly in India, affecting millions. Among diabetic patients, lean type 2 diabetes is a severe subtype with higher microvascular complication risks. While studies on the prevalence, variations and risk factors of diabetes are increasingly available, there has been limited research on the prevalence, variations, and socioeconomic disparities of lean diabetes in India. This study used NFHS-5 microdata, and lean diabetes is defined as those with a BMI level of under 25 and random blood glucose levels of over 200 or under diabetic medication. Descriptive and multivariate analyses were conducted to understand lean diabetes variations and related factors. Socioeconomic disparities were measured using concentration curves and the concentration index. The study unveiled important insights into lean diabetes in India. 8.2% of men and 6.0% of women had elevated blood glucose levels, indicating a significant diabetes burden. Notably, 2.9% of men and 2.4% of women were diagnosed with lean diabetes. Among type 2 diabetics, 52.56% of males and 43.57% of females had lean type 2 diabetes. Lean diabetes prevalence varied from 11.6% in the poorest quintile to 1.1% in the richest. The odds of lean type 2 diabetes among those in the poorest quintile was 6.7 compared to the richest quintile. The concentration index of lean type 2 diabetes was -0.42 for men and -0.39 for women, suggesting a disproportionate impact on lower socioeconomic groups. This study advances our understanding of the complex interplay between socioeconomic factors and lean type 2 diabetes in India. To address the rising burden of lean diabetes among lower socioeconomic strata, policymakers and healthcare professionals must prioritise initiatives enhancing healthcare access, promoting healthy lifestyles, and ensuring effective diabetes management. By addressing socioeconomic disparities and implementing interventions for vulnerable populations, India can reduce diabetes-related mortality and enhance its citizens' overall health.

freely available in the public domain on the official website of DHS. https://dhsprogram.com/data/dataset/India_Standard-DHS_2020.cfm?flag=0.

**Funding:** The authors have not received any funding for this work.

**Competing interests:** The authors have declared that no competing interests exist.

## Background

Globally, the burden of chronic non-communicable diseases (NCDs) is a serious public health concern that jeopardizes social and economic progress [1–3]. Either insufficient insulin production by the pancreas or ineffective insulin uptake by the body causes diabetes, a chronic illness [4–6]. Hyperglycemia, or increased blood sugar, is a common side effect of uncontrolled diabetes that, over time, adversely damages several body systems, including the blood vessels and neurons [7, 8]. The International Diabetes Federation reports that the prevalence of diabetes among adults globally in 2019 was 9.3%, impacting an estimated 463 million individuals [9]. Almost 90% of occurrences of diabetes worldwide are is type 2 diabetes [10, 11]. As of 2019, an estimated 77 million adults in India were estimated to have diabetes, making it a serious public health concern [12]. Type 2 diabetes is more common in urban areas compared to rural areas and among those with higher incomes and levels of education in India than among people with lower socioeconomic status [13].

Although type 2 diabetes is associated with obesity and overweight, a growing body of research indicates that a sizable portion of those with the condition are lean or normal [14]. A sub-type of type 2 diabetes called lean type 2 diabetes has a lower body mass index than typical type 2 diabetes (BMI). People with this sub-type typically have BMIs under 25 kg/m2 [15, 16]. Several factors, such as genetics, ethnicity, and lifestyle choices regarding diet and exercise, can bring this on. Lean type 2 diabetes appears in a more severe form than its counterparts and has a higher risk of micro-vascular consequences [16]. Yet, little is known about the incidence of lean type 2 diabetes and its causes in India. With an emphasis on the socioeconomic and behavioural risk factors contributing to the disease, this study aims to examine the prevalence of lean type 2 diabetes across socio-economic groups in India. Understanding the prevalence and causes of lean type 2 diabetes in India can help us create effective preventive, management, and treatment plans to improve the health of those with the condition. Despite research that used population-based surveys to determine the prevalence and risk factors of diabetes, there hasn't been a reliable estimation for risk factors of lean type 2 diabetes in India.

## Methods

The study utilized the fifth, the latest round of the National Family Health Survey (NFHS-5) carried out by IIPS during 2019–21 [17]. The survey used a multistage stratified sampling technique using the 2011 census to select primary sampling units (PSUs). In the case of rural areas, villages, in urban areas, the Census Enumeration Blocks (CES) were selected as PSU. Details of the survey methodology are available in the national report of the survey [17]. The survey collected information from 636,699 households, 700,564 women in the age group 15–49 years, and 651,946 men in the age group 15–54 years across India. It also provides information on population, health and nutrition, and many important indicators across states/Union Territories for India. Four survey schedules (household, woman, man, and biomarker) were used in local languages using computer-assisted personal interviewing. Among these, the biomarker schedule measured height, weight, waist and hip circumference, blood pressure, and random blood glucose levels for males and females aged 15 years and above. The height of children was measured using an infantometer, while a stadiometer was used for women aged 15–49 and men aged 15–54, and a digital scale was used for weighing. The enumerators measured the random blood glucose using a finger-stick blood specimen for all women and men aged 15 and above using the Accu-Chek Performa glucometer with glucose test strips for blood glucose testing.

### Dependent variable

Impaired blood glucose levels were estimated using random blood glucose levels. Random blood glucose was measured using the ACCU-CHEK machine. A composite variable was computed based on measured random blood glucose, medication and BMI to estimate lean type 2 diabetes. The total sample was classified into five distinct categories. 1) Random blood glucose level <140 without medication, 2) Random blood glucose level <140 with medication, 3) Random blood glucose level between 140–199 without medication, 4) Random blood glucose level between 140–199 with medication, 5) Random blood glucose level 200 and above with or without medication. Further, we narrowed these into three categories for diabetes, **Normal:** Random blood glucose level <140 without medication, **Impaired:** Random blood glucose level between 140–199 without medication, and **Diabetic:** Random blood glucose level ≥ 200 without medication or those patients under medication irrespective of their random blood glucose levels. We defined lean type 2 diabetes as those with a BMI below 25. Further, among the diabetics, we examined the variations with different cut-offs of BMI level (<18.5, <23, and <25) for both males and females. In the age group of 30–49, there were 15,204 females and 3,140 males aged 30–54.

### Independent variables

The socio-economic characteristics used were age group (<20, 20–24, 25–29, 30–34, 35–39, 40–44, 45–49 and 50–54), educational attainment (no education, primary, secondary, and higher), wealth quintile (ranging from poorest to richest), religion (Hindu, Muslims, Christians, Others), social group (categorised as scheduled caste, scheduled tribe, other backward class, and others) marital status (never married, married, and widowed/separated), body mass index (lean, overweight, and obesity), substance use such as alcohol and tobacco (yes/no), and region (north, central, east, northwest, west, and south).

### Statistical analysis

Descriptive statistics were used to measure the prevalence and variations across socio-economic characteristics. Binary logistic regression was carried out to find out the covariates of diabetes among both males and females. The concentration curve and concentration index were used to estimate the socio-economic inequalities in the prevalence of diabetes.

### Concentration curve and concentration index

The concentration curve (CC) and concentration index (CI) have been extensively used to measure health inequality [18–20]. The CC refers to the cumulative proportion of the population based on living conditions (measured by wealth quintile) against the cumulative population of people with diabetes. The CC plots below the equality line show a high concentration of diabetes among the privileged/rich. The CC plot above the line of inequality shows a higher presence of diabetes among the underprivileged/poor group, and the coincidence of the CC plot with the line of equality shows an equitable presence of diabetes across the wealth quintile. CI is derived from CC, whose value ranges from -1 to +1. A '0' value represents uniform distribution [21].

### Findings

Fig 1: Shows the distribution of random blood glucose levels among males (15–54 years) and females (15–49 years). Around 89% were non-diabetic, not under medication, and their random blood glucose level was below 140. About 0.26% had blood glucose levels below 140 but

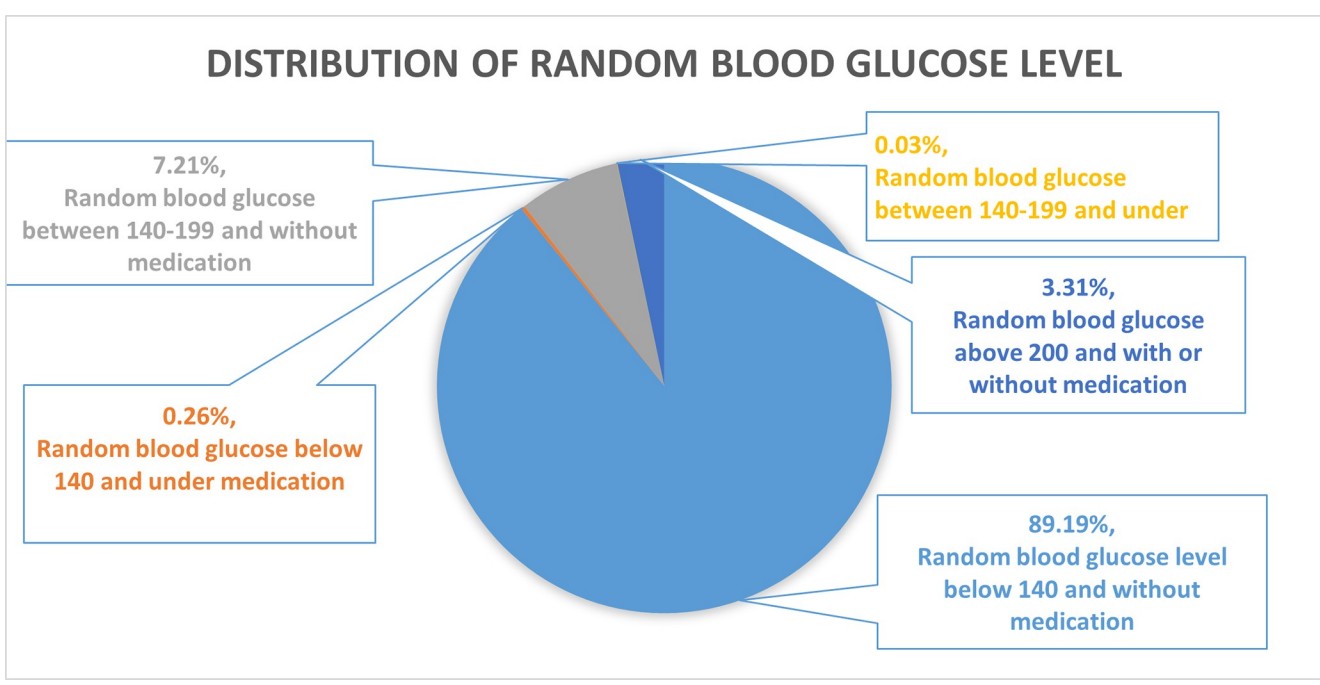

**Fig 1. Distribution of random blood glucose level among females (15–49 years) and males (15–54 years).**

were under medications. For 7.21%, the random blood glucose level ranges from 140–199; they were impaired but not taking medication. Similarly, 0.03% were impaired but taking medication. A total of 3.31% were diabetic, and their blood glucose level was above 200, whether they were under medication or not.

Table 1 presents the prevalence of measured blood glucose among males (15–54 years) and females (15–49 years) by socio-economic characteristics. At the national level, among males, 8.21% had impaired blood glucose levels, and 2.89% had diabetes. The prevalence was lower for females; 6.09% had impaired blood glucose, and 2.39% were diabetic. The prevalence of both impaired blood glucose levels and diabetes increased linearly with age for both males and females. For instance, among males, the estimated impaired blood glucose for those under 20 was 2.79% compared to 15.69% among those in the 50–54 age group. The pattern was similar for females. The impaired and diabetic blood glucose rises among males with no education to higher education. The differences were similar but lower among females. The prevalence of diabetes increased with the wealth quintile. For instance, the prevalence among the poorest females was 1.9% compared to those in the richest, 4.47%. The prevalence was higher among other caste groups and widowed or separated. The prevalence of diabetes was higher among those using tobacco and consuming alcohol compared to their counterparts (S1 Table).

Table 2 presents the percent distribution of diabetes by the varying level of BMI for both males and females. Among all diabetic males (N = 3140), 4.20% were below the BMI level of 18.5 and 26.51% in the BMI level 18–23, 21.85% in the BMI 23–25, and 47.44% above BMI 25. Similarly, among all diabetic females (N = 15204), 3.93% were below the BMI of 18.5, 23.64% were below the BMI level 23, and 16% were in the BMI level of 23–25. Among the males, by BMI <23, 30.71% were diabetic, and it increased to 52.56% by BMI <25. Similarly, for females with BMI <23, 27.57% were diabetic, increasing to 43.57% with BMI <25.

Table 3 presents the distribution of lean diabetes among all type 2 diabetics among males and females by socio-economic characteristics. Diabetes was higher among males compared to females

**Table 1. Prevalence of impaired blood glucose and diabetes among females (15–49 years) and males (15–54 years) by socioeconomic characteristics, India, 2019–21.**

| Variables | Female | | | | Male | | | |
|---|---|---|---|---|---|---|---|---|
| | Normal (RBS<140mg/dl) (%) | Impaired Blood Glucose/ under medication [RBS = 140–199 mg/dl] (%) | Diabetic/ under medication [RBS> = 200 mg/dl] (%) | N | Normal (RBS<140mg/dl) (%) | Impaired Blood Glucose/ under medication [RBS = 140–199 mg/dl] (%) | Diabetic/ under medication [RBS> = 200 mg/dl] (%) | N |
| **Age (in Years)** | | | | | | | | |
| <20 | 96.53 | 2.62 | 0.85 | 1,18,419 | 96.32 | 2.79 | 0.89 | 106641 |
| 20–24 | 95.31 | 3.34 | 1.35 | 1,16,249 | 94.64 | 4.19 | 1.17 | 93442 |
| 25–29 | 93.53 | 4.5 | 1.96 | 1,14,525 | 92.44 | 5.94 | 1.63 | 91274 |
| 30–34 | 90.71 | 6.5 | 2.79 | 97,578 | 88.94 | 8.25 | 2.8 | 83075 |
| 35–39 | 88.22 | 7.74 | 4.05 | 94,327 | 85.48 | 10.03 | 4.48 | 80842 |
| 40–44 | 83.87 | 10.03 | 6.1 | 78,292 | 81.54 | 12.12 | 6.35 | 68800 |
| 45–49 | 80.32 | 11.75 | 7.93 | 81,174 | 77.73 | 13.87 | 8.4 | 70608 |
| 50–54 | | | | | 73.1 | 15.69 | 11.21 | |
| **Education** | | | | | | | | |
| No education | 88.86 | 7.66 | 3.48 | 1,51,234 | 85.38 | 10.95 | 3.67 | 70863 |
| Primary | 88.51 | 7.78 | 3.71 | 88,865 | 85.46 | 10.2 | 4.34 | 79120 |
| Secondary | 91.37 | 5.55 | 3.08 | 3,55,536 | 88.35 | 7.73 | 3.92 | 378283 |
| Higher | 92.36 | 4.72 | 2.92 | 1,04,929 | 88.17 | 7.53 | 4.3 | 123681 |
| **Wealth quintile** | | | | | | | | |
| poorest | 92.02 | 6.1 | 1.88 | 1,32,472 | 89.21 | 8.5 | 2.29 | 115056 |
| poorer | 91.46 | 5.99 | 2.54 | 1,43,210 | 88.79 | 8.25 | 2.96 | 130561 |
| middle | 90.61 | 6.17 | 3.23 | 1,46,140 | 87.85 | 8.21 | 3.94 | 139340 |
| richer | 89.62 | 6.42 | 3.96 | 1,45,555 | 86.66 | 8.35 | 5 | 139317 |
| richest | 89.4 | 6.13 | 4.47 | 1,33,187 | 85.89 | 8.44 | 5.67 | 127672 |
| **Religion** | | | | | | | | |
| Hindu | 90.81 | 6.09 | 3.1 | 5,71,296 | 87.66 | 8.31 | 4.03 | 536318 |
| Muslim | 89.68 | 6.64 | 3.68 | 93,596 | 87.49 | 8.78 | 3.73 | 81446 |
| Christian | 88.05 | 7.05 | 4.9 | 16,831 | 85.04 | 8.86 | 6.09 | 15873 |
| Others | 91.59 | 5.37 | 3.04 | 18,841 | 89.85 | 6.9 | 3.24 | 18308 |
| **Social group** | | | | | | | | |
| SC | 91.09 | 5.83 | 3.08 | 1,57,664 | 88.23 | 8.15 | 3.63 | 144828 |
| ST | 92.67 | 5.51 | 1.82 | 68,317 | 89.93 | 7.64 | 2.43 | 65071 |
| OBC | 90.57 | 6.07 | 3.36 | 2,93,539 | 87.69 | 8.04 | 4.27 | 270720 |
| Others | 89.5 | 6.85 | 3.65 | 1,81,044 | 86.19 | 9.25 | 4.56 | 171327 |
| **Marital status** | | | | | | | | |
| Never married | 95.98 | 2.94 | 1.09 | 1,63,528 | 94.57 | 4.09 | 1.34 | 232142 |
| Married | 89.21 | 7.01 | 3.78 | 5,08,561 | 83.81 | 10.69 | 5.5 | 409915 |
| Widowed/ separate | 84.97 | 9.58 | 5.45 | 28,475 | 83.96 | 10.86 | 5.18 | 9889 |
| **BMI** | | | | | | | | |
| Lean | 92.92 | 4.99 | 2.1 | 5,34,563 | 89.54 | 7.36 | 3.1 | 74336 |
| Overweight | 85.12 | 9.06 | 5.83 | 1,21,145 | 79.87 | 12.79 | 7.34 | 18247 |
| Obese | 77.67 | 12.59 | 9.74 | 43,035 | 73.46 | 14.23 | 12.31 | 3795 |

*(Continued)*

**Table 1.** (Continued)

| Variables | Female | | | | Male | | | |
|---|---|---|---|---|---|---|---|---|
| | Normal (RBS<140mg/dl) (%) | Impaired Blood Glucose/ under medication [RBS = 140–199 mg/dl] (%) | Diabetic/ under medication [RBS>= 200 mg/dl] (%) | N | Normal (RBS<140mg/dl) (%) | Impaired Blood Glucose/ under medication [RBS = 140–199 mg/dl] (%) | Diabetic/ under medication [RBS>= 200 mg/dl] (%) | N |
| **Tobacco** | | | | | | | | |
| No | 90.73 | 6.05 | 3.22 | 6,59,187 | 88.95 | 7.18 | 3.88 | 421334 |
| Yes | 88.79 | 7.97 | 3.24 | 41,377 | 85.25 | 10.48 | 4.27 | 230612 |
| **Alcohol** | | | | | | | | |
| No | 90.62 | 6.15 | 3.23 | 6,93,764 | 88.36 | 7.86 | 3.78 | 530446 |
| Yes | 90.39 | 7.25 | 2.37 | 6,800 | 84.48 | 10.47 | 5.05 | 121500 |
| **Region** | | | | | | | | |
| North | 93.43 | 3.99 | 2.58 | 97,588 | 91.22 | 5.83 | 2.95 | 90973 |
| Central | 92.67 | 4.88 | 2.45 | 1,70,127 | 90.4 | 6.65 | 2.95 | 155321 |
| East | 88.87 | 8.09 | 3.04 | 1,63,797 | 84.72 | 11.23 | 4.05 | 138481 |
| Northeast | 89.74 | 7.31 | 2.95 | 26,749 | 86.01 | 10.1 | 3.89 | 25919 |
| West | 91.52 | 6.09 | 2.39 | 98,372 | 88.91 | 8.21 | 2.89 | 106973 |
| South | 87.81 | 6.8 | 5.38 | 1,43,932 | 84.34 | 8.8 | 6.86 | 134280 |
| **India** | 90.62 | 6.16 | 3.22 | 7,00,564 | 87.64 | 8.34 | 4.02 | 6,51,946 |

Note: Person file is used. A person is called diabetic if their glucose level is ≥ 200 or if they are under medication

across the BMI level. The percentage of lean type 2 diabetes increased with increasing BMI cutoffs but decreased with increasing age. For instance, among males with BMI <18.5 and aged 30–39, 6.72% were diabetic compared to 3.28% among the 40+ age group. The pattern remained similar for females, with a lower prevalence rate across all the BMI cutoffs. Further, the distribution of lean type 2 diabetes decreased linearly with the increasing level of education and wealth quintile across the BMI cutoffs for both males and females. Among those with a BMI below 18.5, it ranged from 9.16% among males with no education to 1.85% among those with higher education. By wealth quintile, it ranged from 11.5% among the poorest to 2.36% among the richest, with a similar pattern among females. Similarly, by the social group, the prevalence was higher among the marginalised groups like SC and ST population; for instance, type 2 diabetes among males was 9.05% among STs and 1.52% among other social groups with BMI below 18.5. The prevalence increased with higher BMI cutoffs among both males and females. Additionally, the distribution of type 2 lean diabetes was higher among those consuming alcohol and tobacco compared to those who don't. The gap was more evident among females compared to males.

**Table 2.** Among all type 2 diabetic females aged 30–49 years and males aged 30–54 years, the percent distribution of diabetic cases by BMI level in India, 2019–21.

| BMI | Diabetes | |
|---|---|---|
| | Female% (95% CI) | Male% (95% CI) |
| <18.5 | 3.93 (3.57–4.32) | 4.20 (3.33–5.29) |
| 18.5–23 | 23.64 (22.74–24.56) | 26.51 (24.48–28.65) |
| 23–25 | 16 (15.21–16.81) | 21.85 (19.89–23.95) |
| >25 | 56.43 (55.34–57.52) | 47.44 (45.01–49.89) |
| N | 15,204 | 3,140 |

Note: Categorisation of BMI cutoffs is continuous in nature.

**Table 3. Distribution of lean diabetes among all type 2 diabetes cases among females (30–49 years) and males (30–54 years) by socio-demographic characteristics, India, 2019–21.**

| Variables | Female | | | | Male | | | |
|---|---|---|---|---|---|---|---|---|
| | BMI <18.5 | BMI <23 | BMI <25 | N | BMI <18.5 | BMI <23 | BMI <25 | N |
| **Age (in Years)** | | | | | | | | |
| 30–39 | 5.4 (4.74, 6.13) | 31.75(30.17, 33.36) | 47.45 (45.7, 49.21) | 5,605 | 6.72 (4.74, 9.44) | 35.52 (31.33, 39.94) | 55.22 (50.59, 59.77) | 840 |
| 40+ | 3.08 (2.67, 3.54) | 25.13 (23.96, 26.34) | 41.30 (39.92, 42.69) | 9,599 | 3.28 (2.40, 4.47) | 28.95 (26.48, 31.56) | 51.59 (48.7, 54.46) | 2300 |
| **Education** | | | | | | | | |
| No education | 6.09 (5.31, 6.98) | 36.65 (34.8, 38.53) | 53.58 (51.59, 55.55) | 4,190 | 9.16 (5.5, 14.89) | 49.46 (42.03, 56.91) | 71.20 (64.24, 77.29) | 326 |
| Primary | 4.95 (3.93, 6.22) | 29.81 (27.45, 32.28) | 46.83 (44.13, 49.54) | 2,510 | 7.97 (5.09, 12.26) | 35.87 (30.06, 42.12) | 61.63 (55.47, 67.44) | 430 |
| Secondary | 2.82 (2.39, 3.33) | 23.18 (21.83, 24.58) | 38.54 (36.94, 40.15) | 6,926 | 3.2 (2.22, 4.58) | 28.93 (26.15, 31.87) | 49.88 (46.67, 53.10) | 1758 |
| Higher | 1.44 (0.94, 2.2) | 19.18 (16.65, 22.01) | 33.88 (30.53, 37.39) | 1,579 | 1.85 (1.01, 3.37) | 22.41 (18.27, 27.17) | 44.13 (38.53, 49.89) | 626 |
| **Wealth Quintile** | | | | | | | | |
| poorest | 11.61 (9.77, 13.74) | 54.62 (51.41, 57.78) | 73.12 (70.18, 75.87) | 1,548 | 11.5 (7.24, 17.78) | 53.89 (46.09, 61.49) | 74.96 (67.52, 81.17) | 298 |
| poorer | 6.84 (5.77, 8.09) | 41.36 (38.8, 43.97) | 59.59 (56.91, 62.22) | 2,302 | 8.96 (5.82, 13.56) | 46.06 (40.09, 52.15) | 67.66 (61.99, 72.86) | 416 |
| middle | 4.12 (3.36, 5.04) | 29.61 (27.52, 31.78) | 45.64 (43.28, 48.02) | 3,083 | 3.84 (2.46, 5.94) | 31.37 (26.89, 36.22) | 54.56 (49.37, 59.65) | 659 |
| richer | 2.15 (1.62, 2.83) | 21.5 (19.80, 23.32) | 37.62 (35.59, 39.71) | 4,033 | 2.36 (1.39, 3.97) | 30.09 (26.06, 34.45) | 51.87 (47.33, 56.38) | 857 |
| richest | 1.1 (0.8, 1.51) | 14.48 (13.05, 16.03) | 28.20 (26.27, 30.22) | 4,237 | 1.63 (0.67, 3.94) | 16.20 (13.13, 19.82) | 37.51 (32.94, 42.31) | 910 |
| **Religion** | | | | | | | | |
| Hindu | 4.05 (3.65, 4.48) | 28.39 (27.33, 29.49) | 44.57 (43.34, 45.81) | 11,891 | 4.59 (3.59, 5.84) | 31.01 (28.64, 33.48) | 52.87 (50.15, 55.58) | 2577 |
| Muslim | 3.49 (2.57, 4.71) | 25.33 (22.85, 27.99) | 40.02 (37.12, 42.98) | 2,325 | 1.26 (0.36, 4.34) | 29.36 (23.13, 36.47) | 50.86 (43.60, 58.08) | 369 |
| Christian | 4.33 (2.76, 6.73) | 23.98 (19.90, 28.60) | 42.65 (37.46, 48.01) | 595 | 5.30 (1.40, 18.05) | 33.92 (23.05, 46.81) | 54.58 (42.50, 66.14) | 108 |
| Others | 2.49 (1.32, 4.63) | 21.24 (16.92, 26.3) | 35.57 (30.61, 40.87) | 393 | 3.83 (1.34, 10.47) | 23.55 (15.11, 34.79) | 47.87 (36.57, 59.39) | 86 |
| **Social group** | | | | | | | | |
| SC | 4.88 (4.03, 5.9) | 31.61 (29.47, 33.83) | 47.91 (45.54, 50.29) | 3,256 | 6.53 (4.23, 9.95) | 38.79 (33.69, 44.14) | 61.77 (56.36, 66.9) | 580 |
| ST | 8.43 (6.73, 10.51) | 40.96 (37.05, 44.98) | 59.09 (54.83, 63.21) | 767 | 9.05 (5.64, 14.22) | 49.81 (41.76, 57.87) | 64.06 (56.01, 71.39) | 180 |
| OBC | 3.99 (3.46, 4.61) | 27.80 (26.43, 29.21) | 43.82 (42.25, 45.39) | 6,582 | 4.35 (3.02, 6.21) | 28.12 (25.15, 31.28) | 51.36 (47.87, 54.84) | 1468 |
| Others | 2.42 (1.93, 3.03) | 22.15 (20.45, 23.94) | 37.54 (35.46, 39.66) | 4,599 | 1.52 (0.82, 2.80) | 25.97 (21.97, 30.43) | 46.35 (41.49, 51.28) | 912 |
| **Marital status** | | | | | | | | |
| Never married | 6.35 (3.38, 11.6) | 35.12 (27.73, 43.3) | 51.60 (42.91, 60.18) | 247 | 8.36 (3.85, 17.21) | 42.04 (30.80, 54.17) | 60.76 (48.28, 71.97) | 102 |
| Married | 3.87 (3.5, 4.28) | 27.37 (26.38, 28.39) | 43.44 (42.29, 44.59) | 13,686 | 4.05 (3.16, 5.17) | 30.16 (27.95, 32.46) | 52.19 (49.66, 54.7) | 2981 |
| Widowed/separated | 4.14 (3.02, 5.66) | 28.21 (24.98, 31.67) | 43.40 (39.71, 47.17) | 1,271 | 4.66 (1.38, 14.61) | 39.35 (25.52, 55.13) | 57.36 (41.19, 72.11) | 57 |
| **Tobacco** | | | | | | | | |
| No | 3.59 (3.23, 3.98) | 26.41 (25.44, 27.4) | 42.37 (41.24, 43.5) | 14,167 | 2.44 (1.69, 3.50) | 25.18 (22.61, 27.93) | 47.61 (44.46, 50.78) | 1846 |
| Yes | 8.6 (6.84, 10.77) | 43.46 (39.62, 47.38) | 59.94 (56.04, 63.73) | 1,037 | 6.72 (5.00, 8.97) | 38.60 (35.01, 42.32) | 59.62 (55.80, 63.32) | 1294 |
| **Alcohol** | | | | | | | | |
| No | 3.89 (3.53, 4.28) | 27.47 (26.52, 28.44) | 43.47 (42.38, 44.57) | 15,087 | 3.31 (2.44, 4.46) | 29.22 (26.74, 31.82) | 51.32 (48.45, 54.18) | 2350 |
| Yes | 9.24 (5.62, 14.82) | 40.92 (32.30, 50.13) | 55.76 (46.15, 64.95) | 117 | 6.85 (4.77, 9.76) | 35.15 (30.93, 39.62) | 56.25 (51.58, 60.81) | 790 |
| **Region** | | | | | | | | |
| North | 2.9 (2.22, 3.79) | 22.85 (21.02, 24.80) | 37.92 (35.78, 40.11) | 1,670 | 3.48 (1.95, 6.12) | 24.26 (20.02, 29.07) | 47.55 (42.26, 52.90) | 329 |
| Central | 5.33 (4.48, 6.33) | 34.62 (32.57, 36.74) | 51.49 (49.27, 53.71) | 2,599 | 5.83 (3.86, 8.74) | 35.85 (31.16, 40.81) | 56.93 (51.74, 61.98) | 482 |
| East | 5.74 (4.76, 6.91) | 35.74 (33.33, 38.22) | 51.85 (49.26, 54.43) | 3,326 | 4.56 (2.82, 7.30) | 38.39 (33.21, 43.86) | 64.96 (59.71, 69.88) | 703 |
| Northeast | 7.19 (5.59, 9.21) | 40.15 (36.93, 43.45) | 59.63 (56.36, 62.82) | 500 | 2.50 (1.19, 5.17) | 31.13 (25.01, 38.00) | 59.70 (52.56, 66.45) | 120 |
| West | 3.49 (2.55, 4.76) | 25.32 (21.80, 29.19) | 40.19 (35.83, 44.71) | 1,567 | 1.73 (0.86, 3.45) | 22.86 (17.30, 29.56) | 39.33 (31.45, 47.8) | 384 |
| South | 2.33 (1.87, 2.9) | 20.28 (18.89, 21.75) | 36.08 (34.34, 37.87) | 5,542 | 4.51 (2.90, 6.95) | 28.22 (24.59, 32.16) | 48.14 (43.96, 52.35) | 1122 |

Table 4 shows the occurrence of lean type 2 diabetes among males and females, controlling for socio-economic variables. The adjusted odds ratio (AOR) of lean type 2 diabetes was higher among those with no or a low education level, those belonging to the poorest and poorer

**Table 4. Association of lean type 2 diabetes with socioeconomic variables among females (30–49 years) and males (30–54 years) India, 2019–21, dependent variable (0 = no diabetic 1 diabetic).**

| Variables | Adjusted odds Ratio with 95% CI | | |
|---|---|---|---|
| **Sex** | **BMI <18.5** | **BMI <23** | **BMI <25** |
| Female | 1.00 | 1.00 | 1.00 |
| Male | 0.93 (0.67, 1.28) | 1.13 (0.97, 1.3) | 1.46***(1.28, 1.67) |
| **Age (in Years)** | | | |
| 40+ | 1.00 | 1.00 | 1.00 |
| 30–39 | 1.81***(1.49, 2.19) | 1.34***(1.22, 1.48) | 1.22***(1.11, 1.33) |
| **Education** | | | |
| Higher | 1.00 | 1.00 | 1.00 |
| Secondary | 1.27 (0.84, 1.92) | 0.97 (0.82, 1.15) | 0.97 (0.84, 1.13) |
| Primary | 1.76**(1.11, 2.78) | 1.02 (0.84, 1.24) | 1.08 (0.90, 1.29) |
| No education | 1.73**(1.10, 2.72) | 1.19*(0.98, 1.44) | 1.19*(1.00, 1.41) |
| **Wealth Quintile** | | | |
| Richest | 1.00 | 1.00 | 1.00 |
| Richer | 1.61**(1.05, 2.47) | 1.66***(1.43, 1.93) | 1.53***(1.35, 1.73) |
| Middle | 2.85***(1.88, 4.34) | 2.27***(1.94, 2.65) | 1.95***(1.70, 2.23) |
| Poorer | 4.38***(2.84, 6.75) | 3.42***(2.90, 4.04) | 3.06***(2.63, 3.56) |
| Poorest | 6.66***(4.17, 10.63) | 5.06***(4.17, 6.14) | 4.94***(4.10, 5.96) |
| **Religion** | | | |
| Muslim | 1.00 | 1.00 | 1.00 |
| Hindu | 1.28 (0.90, 1.82) | 1.22***(1.06, 1.41) | 1.28***(1.13, 1.46) |
| Christian | 1.86**(1.04, 3.33) | 1.35**(1.02, 1.78) | 1.56***(1.22, 2.01) |
| Others | 1.05 (0.54, 2.05) | 1 (0.74, 1.36) | 1.08 (0.84, 1.39) |
| **Social group** | | | |
| Others | 1.00 | 1.00 | 1.00 |
| SC | 1.51**(1.09, 2.08) | 1.28***(1.11, 1.48) | 1.24***(1.08, 1.41) |
| ST | 1.89***(1.33, 2.68) | 1.53***(1.26, 1.86) | 1.45***(1.20, 1.75) |
| OBC | 1.71***(1.29, 2.25) | 1.33***(1.18, 1.50) | 1.31***(1.18, 1.46) |
| **Marital status** | | | |
| Married | 1.00 | 1.00 | 1.00 |
| Never married | 1.9**(1.09, 3.28) | 1.47**(1.08, 2.00) | 1.34*(0.98, 1.83) |
| Widowed/separated | 1.04 (0.73, 1.46) | 1 (0.84, 1.20) | 0.93 (0.79, 1.10) |
| **Tobacco** | | | |
| No | 1.00 | 1.00 | 1.00 |
| Yes | 1.58***(1.20, 2.08) | 1.36***(1.17, 1.59) | 1.26***(1.09, 1.46) |
| **Alcohol** | | | |
| No | 1.00 | 1.00 | 1.00 |
| Yes | 1.33 (0.88, 2.02) | 1.03 (0.83, 1.29) | 0.98 (0.80, 1.22) |
| **Region** | | | |
| Northeast | 1.00 | 1.00 | 1.00 |
| North | 1.11 (0.73, 1.70) | 0.93 (0.77, 1.12) | 0.78***(0.66, 0.93) |
| Central | 0.96 (0.66, 1.39) | 0.96 (0.81, 1.14) | 0.82**(0.69, 0.97) |
| East | 0.94 (0.66, 1.34) | 0.94 (0.79, 1.12) | 0.8**(0.68, 0.95) |
| West | 1.06 (0.68, 1.66) | 0.94 (0.75, 1.18) | 0.74***(0.60, 0.92) |

(*Continued*)

**Table 4.** (Continued)

| Variables | Adjusted odds Ratio with 95% CI | | |
|---|---|---|---|
| **Sex** | **BMI <18.5** | **BMI <23** | **BMI <25** |
| South | 0.72 (0.49, 1.07) | 0.65***(0.55, 0.78) | 0.58***(0.49, 0.68) |

Note

*if p < 0.05

**if p < 0.01

***if p < 0.001; CI: Confidence Interval. All the AOR are adjusted for background characteristics like age, sex, wealth, quintile level of education, etc.

wealth quintile, and lower social strata like SCs and STs. The AOR was higher among males than females and those using any substance like alcohol or tobacco. The AOR of lean type 2 diabetes varied with varying cutoffs of BMI. For instance, with a BMI <25, the odds of lean type 2 diabetes among males were 1.46 (95% CI: 1.28–1.67) compared to females. Further, the odds of lean diabetes among those with no education were 1.19 (95% CI: 1.00–1.41) compared to those with higher education; similarly, the odds were 4.94 (95% CI: 4.10–5.96) among the poorest wealth quintile compared to the richest. In the case of the social group, the odds were 1.45 (95% CI: 1.20–1.46) among STs than the others. The pattern was similar among those with BMI <18.5 and BMI <23 [Fig 2A–2C].

Table 5 presents the concentration index by three cutoff points of BMI for males and females with a p-value. Overall values of the index are negative, indicating a higher concentration among the poor. For BMI <18.5, the concentration index was -0.42 (95% CI: -045–0.39) among males and -0.39 (95% CI: -0.47–0.31) among females. Similarly, for those with BMI <23, the concentration index was -0.25 (95% CI: -26–0.24) for males and -0.21 (-0.23–0.19) for females. The pattern remains similar for males and females, with a comparatively lower index value of BMI <25. Table 6 shows the concentration index for impaired blood glucose levels (140–199 mg/dl). Likewise, for impaired blood glucose, the value of the index is negative, showing a higher concentration among the poor. Also, the pattern remains similar with different BMI cutoffs.

Fig 3 shows the concentration curve of type 2 lean diabetes among males and females with different BMI cutoffs. For BMI <18.5, the concentration curve was above the line of equality, showing a higher concentration of type 2 lean diabetes among low-income people. This pattern holds true for other cutoff points: BMI <23 and BMI <25.

Fig 4 shows the state-wise distribution of lean type 2 diabetes in India. The prevalence of lean type 2 diabetes was higher among the relatively less developed states like Jharkhand, followed by Rajasthan and Madhya Pradesh. The prevalence was comparatively low among Delhi, followed by Punjab and Andhra Pradesh (S2 Table).

## Discussion

Diabetes is a significant public health concern in India, with an estimated 77 million adults living with the disease as of 2019 [11]. Among all diabetic cases, lean type 2 diabetes has proven to have a poor prognosis compared to other counterparts. Studies also suggest that lean diabetes has increased total, cardiovascular, and non-cardiovascular mortality when compared to obese diabetic patients [16]. This is the first-ever study that examined the socio-economic variations of type 2 lean diabetes in India using nationally representative data among females aged 30–49 and males aged 30–54. Our salient findings are as follows.

A

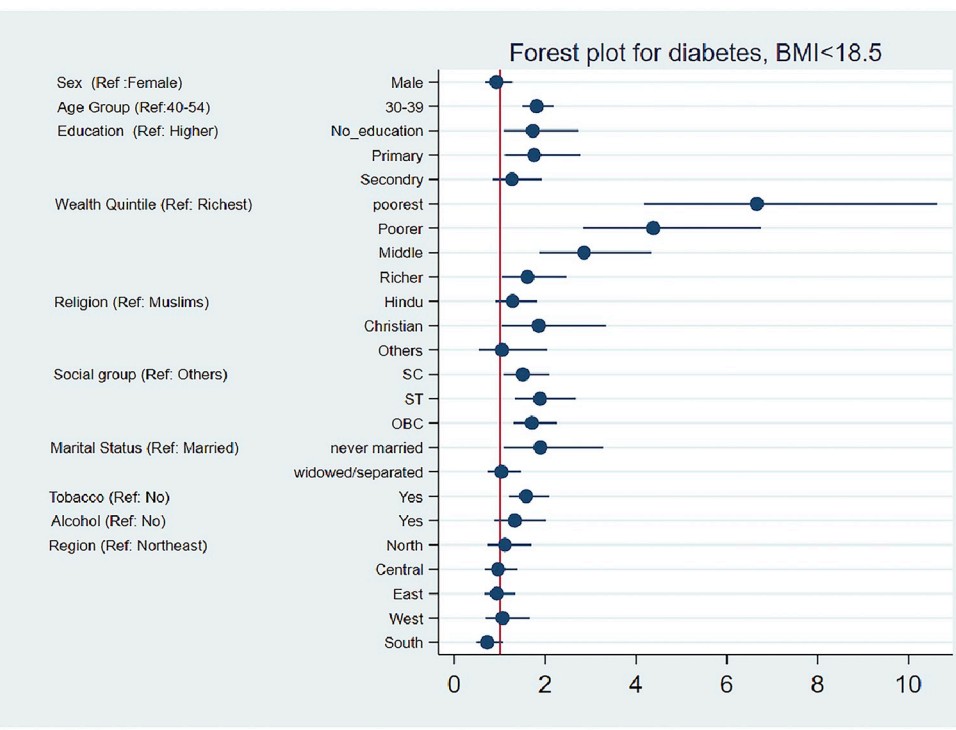

B

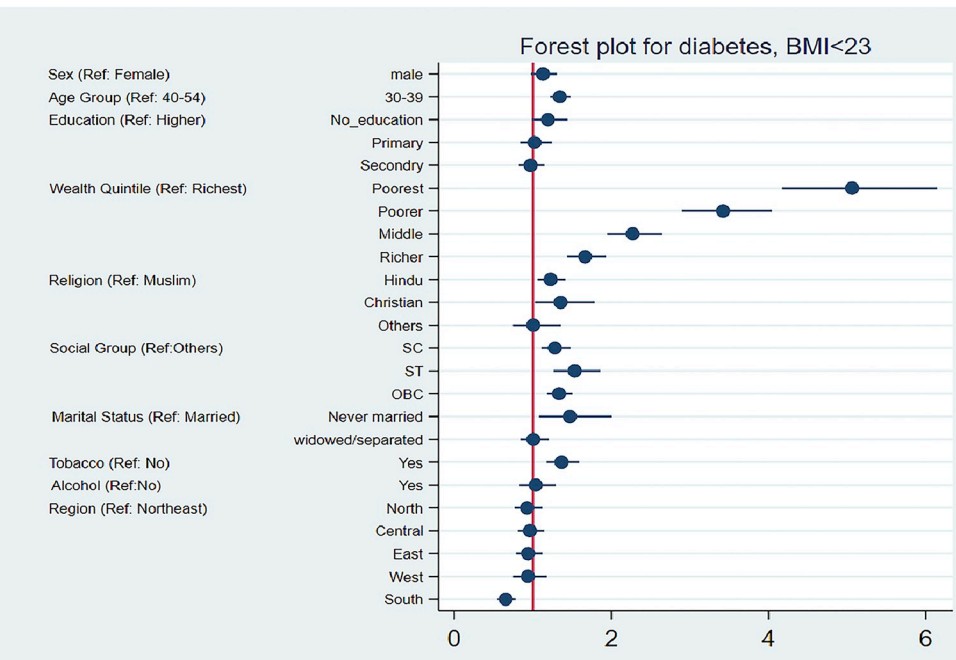

**Fig 2.** Forest plot of lean type 2 diabetes by BMI cutoff A) <18.5, B) <23, and C) <25 with socioeconomic variables among females (30–49 years) and males (30–54 years), India, 2019–21.

**Table 5. Concentration index of lean type 2 diabetic among females (30–49 years) and males (30–54 years), India, 2019–21.**

|  | Female | P Value | Male | P Value |
|---|---|---|---|---|
| BMI | Concentration Index (95% CI) |  | Concentration Index (95% CI) |  |
| <18.5 | -0.42 (-0.45–0.39) | 0 | -0.39 (-0.47–0.31) | 0 |
| <23 | -0.25 (-0.26–0.24) | 0 | -0.21 (-0.23–0.19) | 0 |
| <25 | -0.18 (-0.18–0.17) | 0 | -0.12 (-0.14–0.11) | 0 |

**Table 6. Concentration index of impaired blood glucose level (140–199 mg/dl) among females (30–49 years) and males (30–54 years), India, 2019–21.**

|  | Female | P Value | Male | P Value |
|---|---|---|---|---|
| BMI | Concentration Index (95% CI) |  | Concentration Index (95% CI) |  |
| <18.5 | -0.45( -0.48 - 0.42) | 0 | -0.43( -0.49 - 0.35) | 0 |
| <23 | -0.28( -0.30 - 0.26) | 0 | -0.24( -0.27 - 0.21) | 0 |
| <25 | -0.20( -0.23 - 0.18) | 0 | -0.14( -0.17 - 0.12) | 0 |

First, the overall prevalence of type 2 diabetes increases with age, was higher among males, higher in urban areas, and higher among obese. In contrast, the prevalence of lean type 2 diabetes increases with age, higher among males and higher in rural areas (S1 Table). Multivariate analyses of lean diabetes among diabetic samples confirmed age, sex, residence and behavioural factors as significant risk factors. Second, regional variation in lean diabetes is large. Among all diabetic cases, lean diabetes was higher in poorer states of India.

For instance, the highest prevalence of lean diabetes was observed in Bihar, followed by Uttar Pradesh, Odisha, Jharkhand, Madhya Pradesh, and Assam. Third, the prevalence of type 2 diabetes in India varies by socioeconomic status, with higher among individuals with higher income (wealth quintile), education levels and those from non SC/ST. In contrast, we found a higher prevalence of lean type 2 diabetes among those with low or no education and marginalized groups (SCs and STs), suggesting that interventions are needed to address these disparities. Additionally, the higher prevalence of lean type 2 diabetes among those using substances like alcohol and tobacco highlights the need for targeted interventions to reduce substance use and improve overall health outcomes. Fourth, among those who are diabetic, the concentration index of type 2 lean diabetes and pre-diabetic is pro-poor, suggesting a larger concentration among poor and disadvantageous for both males and females.

Our findings on estimates and risk factors of diabetes and type 2 lean diabetes are consistent with the literature [22–24]. A small-scale community-based study from the state of Tamil Nadu, India, by Oommen et al. found that the chance of lean diabetes was higher among older, illiterate, and those involved in manual labour compared to those with non-lean diabetes [25]. A study with longer follow-ups from Germany have estimated that lean diabetes has a 2.52 times higher odds ratio (OR) for mortality and 2.50 times higher OR for hypoglycemia compared to diabetes with a BMI subgroup $\geq$30 kg/m2 [26]. Literature also suggests that there are significant wealth-based and education-based inequalities in the prevalence and management of hypertension in India and Nepal [27]. Similarly, undiagnosed hypertension was disproportionately higher among lower socioeconomic status groups (Concentration Index, C = −0.18) in Nepal [28].

We provide some plausible explanations of the significant socioeconomic gradient in lean diabetes and pre-diabetes prevalence in India. The potential causal mechanisms of lean diabetes and pre-diabetes may include genetic, acquired and behavioural factors. Evidence from

A

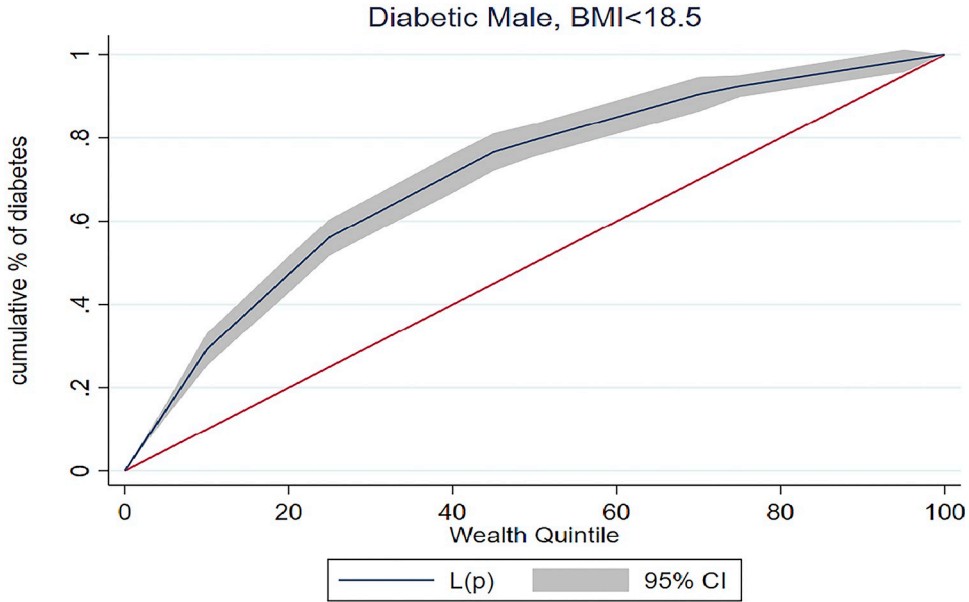

B

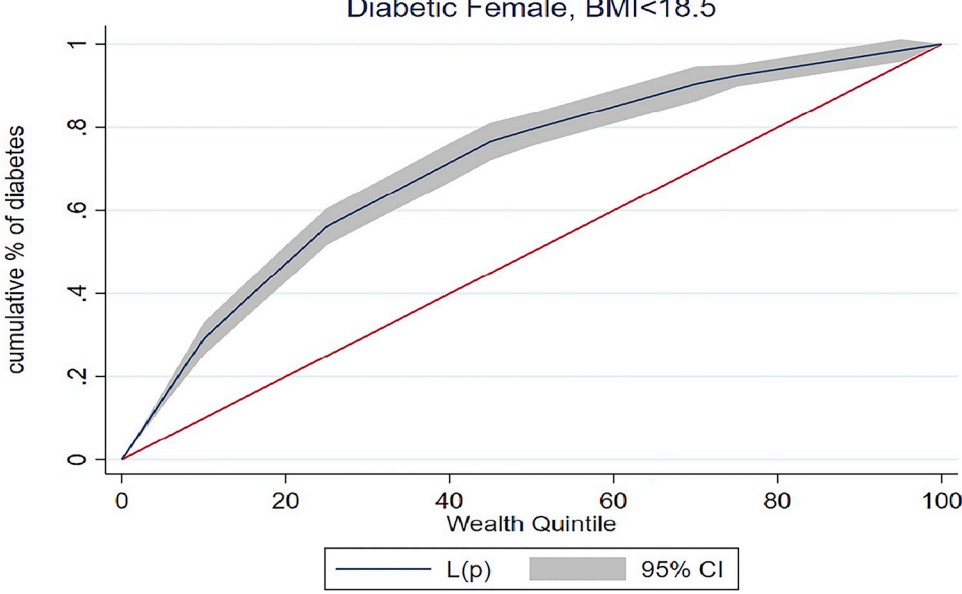

**Fig 3. Concentration curve of lean type 2 diabetic among females (30–49 years) and males (30–54 years), India, 2019–21.**

developing countries suggests that lean diabetes patients include a history of childhood malnutrition, poor socioeconomic status, and a relatively early age of onset [16]. Besides, sedentary lifestyle, dietary practices, stress, and financial impoverishment may aggravate diabetic conditions [16, 29, 30].

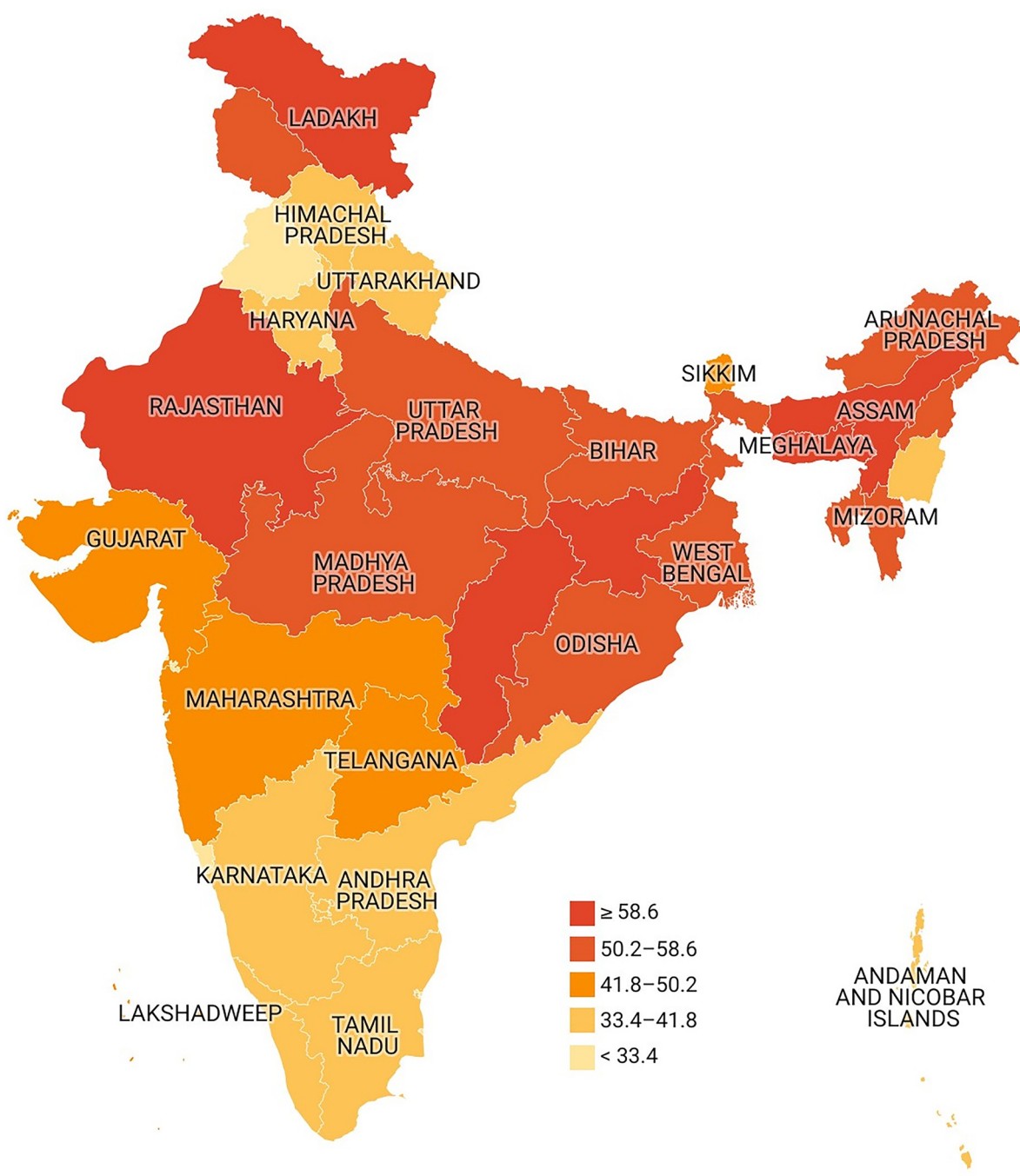

**Fig 4. Represents the state-wise distribution of lean type 2 diabetes in India.** (https://github.com/datameet/maps). (Authors generated the estimates employing the NFHS 5 data).

Chronic diseases like diabetes and hypertension are no more the diseases of rich and obese people. The fact that type 2 diabetes can develop in individuals with normal or low BMI underscores the importance of regular screening and early detection of diabetes, particularly in high-risk populations. It also highlights the need for tailored treatment approaches for lean diabetes, which may differ from those for diabetes brought on by obesity. Community awareness related to lean diabetes and pre-diabetes need to be emphasized. Public health measures

are required to control blood sugar among lean type 2 diabetes and prevent the progression of lean pre-diabetes to lean diabetes as they have a higher rate of complications and are more common among those lower wealth quintiles who may not afford the treatment. Universal health care can help to reduce health disparities related to chronic health conditions, including diabetes, by ensuring all individuals, regardless of socio-economic status, have access to the same high-quality care and resources for managing their condition. Universal health care contributes to better health outcomes and quality of life with access to essential health services, promoting preventive care, emphasizing chronic disease management, and reducing health disparities associated with lean pre-diabetes and lean diabetes.

The study has the following limitations. We could not conduct district-level analysis as the sample size for district-level analyses was inadequate. Second, the measurement of random blood sugar was taken in the field setting and not validated with HbA1C.

Despite these limitations, we assert that the conventional protocols for managing type 2 diabetes may not be suitable for individuals with 'lean diabetes' residing in impoverished rural communities in developing nations. It is crucial to enhance comprehension and management of lean diabetes, suggesting the development of effective care plans and delivering suitable treatment for these marginalized groups of patients. Thus, lean pre-diabetes and diabetes need a multi-sectoral approach for prevention and management, including addressing socioeconomic determinants, promoting healthy lifestyles and behaviors, and improving access to healthcare and diabetes screening.

## Supporting information

**S1 Table. Prevalence of lean diabetes among females (15–49 years) and males (15–54) by socio-demographic characteristics, India, 2019–21.**
(DOCX)

**S2 Table. State pattern of lean type 2 diabetes, India, 2019–21.**
(DOCX)

## Author Contributions

**Conceptualization:** Surama Manjari Behera, Priyamadhaba Behera, Sanjay K. Mohanty.

**Data curation:** Sanjay K. Mohanty, Rajeev Ranjan Singh.

**Formal analysis:** Rajeev Ranjan Singh.

**Methodology:** Sanjay K. Mohanty, Rajeev Ranjan Singh.

**Software:** Rajeev Ranjan Singh.

**Writing – original draft:** Surama Manjari Behera, Rajeev Ranjan Singh, Avinaba Mukherjee.

**Writing – review & editing:** Surama Manjari Behera, Priyamadhaba Behera, Sanjay K. Mohanty, Rajeev Ranjan Singh, Binod Kumar Patro, Avinaba Mukherjee, Venkatarao Epari.

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
