## [Decision Letter · Decision Letter 0]

12 Feb 2024

PGPH-D-23-02099

Socioeconomic Gradient of Lean Diabetes in India: Evidence from National Family Health Survey, 2019-21

Dear Dr. Ranjan Singh,

Thank you for submitting your manuscript to PLOS Global Public Health. After careful consideration, we feel that it has merit but does not fully meet PLOS Global Public Health’s publication criteria as it currently stands. Therefore, we invite you to submit a revised version of the manuscript that addresses the points raised during the review process.

EDITOR: Dear Author,

Please attend to all the comments provided by the reviewers and make the necessary corrections. 

The decision of this manuscript is justified based on PLOS Global Public Health’s publication criteria and not on its novelty or perceived impact.

We look forward to receiving your revised manuscript.

Kind regards,

Zulkarnain Jaafar

Academic Editor

Journal Requirements:

- https://doi.org/10.1186/s12909-022-03162-8

- https://doi.org/10.1016/j.acalib.2019.02.006

In your revision ensure you cite all your sources (including your own works), and quote or rephrase any duplicated text outside the methods section. Further consideration is dependent on these concerns being addressed.

3. Please send a completed 'Competing Interests' statement, including any COIs declared by your co-authors. If you have no competing interests to declare, please state "The authors have declared that no competing interests exist". Otherwise please declare all competing interests beginning with the statement "I have read the journal's policy and the authors of this manuscript have the following competing interests:"

5. Please provide separate figure files in .tif or .eps format only and remove any figures embedded in your manuscript file. Please also ensure all files are under our size limit of 10MB.

6. Some material included in your submission may be copyrighted. According to PLOS’s copyright policy, authors who use figures or other material (e.g., graphics, clipart, maps) from another author or copyright holder must demonstrate or obtain permission to publish this material under the Creative Commons Attribution 4.0 International (CC BY 4.0) License used by PLOS journals. Please closely review the details of PLOS’s copyright requirements here: PLOS Licenses and Copyright. If you need to request permissions from a copyright holder, you may use PLOS's Copyright Content Permission form.

Potential Copyright Issues:

Fig 1: please (a) provide a direct link to the base layer of the map (i.e., the country or region border shape) and ensure this is also included in the figure legend; and (b) provide a link to the terms of use / license information for the base layer image or shapefile. We cannot publish proprietary or copyrighted maps (e.g. Google Maps, Mapquest) and the terms of use for your map base layer must be compatible with our CC-BY 4.0 license. 

"

Additional Editor Comments (if provided):

Reviewers' comments:

Reviewer's Responses to Questions

**Comments to the Author**

1. Does this manuscript meet PLOS Global Public Health’s publication criteria? Is the manuscript technically sound, and do the data support the conclusions? The manuscript must describe methodologically and ethically rigorous research with conclusions that are appropriately drawn based on the data presented.

Reviewer #1: Yes

Reviewer #2: Yes

2. Has the statistical analysis been performed appropriately and rigorously?

Reviewer #1: Yes

Reviewer #2: Yes

3. Have the authors made all data underlying the findings in their manuscript fully available (please refer to the Data Availability Statement at the start of the manuscript PDF file)?

Reviewer #1: Yes

Reviewer #2: Yes

4. Is the manuscript presented in an intelligible fashion and written in standard English?

Reviewer #1: Yes

Reviewer #2: Yes

5. Review Comments to the Author

Reviewer #1: Dear Authors,

It is an important scientific inquiry that focuses out of the conventional inquiry of diabetes, that is lean DM, which is important for the communities with low BMI (less than overweight), and so, is an important issue in public health.

The most literature (more than 50%) have been found within recent within 5 years, written in comprehensible way, and with rational justified. With suffice sample, the data analysis has been carried out in accordance with scientific rigour. Descriptive analysis along with logistic regression, he concentration index with their 95% CI, have been reported. The figures added further understanding of the DM distribution and inequality variations among different populations segments. However, the following are commented on:

1. Despite having analysis and results reported clearly, the discussion looks weaker in comparison. Although lean DM and its associated factors have been discussed clearly, its pre/sub-clinical situation, which might be important from pathogenetic dimensions and thus screened earlier, impacting higher cost-effectiveness, and even more important from the UHC. Taken as example, the pre-HTN and pre-clinical scenario have been missed, as these have been found with opposite relationship with the obvious HTN and such other chronic diseases. When the HTN was found to be higher among the richer segments, the Pre-HTN was significantly higher with C~-0.18 among the poor and, how these would impact the UHC, has been clearly explored (cited in our review article: Adhikari et al. Need for HTA supported risk factor screening for hypertension and diabetes in Nepal: A systematic scoping review). This is further supported as in your results that the inequalities (C values) towards higher BMI are further decreased, as compared to the lower BMIs. This may indicate that the pre-DM situation might be higher among with the lower BMIs, even lower than 18.5, which may coincide and be a fate of the poor. So, I recommend adding further discussion regarding the pre-DM scenario with the values of C in India, and even among the countries with similar socio-economic statuses.

2. Logistic regression is seemed to have been calculated only for raw odds ratios. These could be further calculated for adjustment, which may add further importance of associated factors, with further determination.

3. Concentration indexes have been reported with 95% CI though, it would be better to further add the p-values, which may add easiness in interpretations for the readers.

4. Finally, Some words are found joined together, however, this may have been gotten during the pdf conversion process, still hopefully, attentions would be given.

It is further hypothesized with undiagnosed DM, which is missed, in addition to the Pre-DM, that is mentioned. So, hope to be discussed in that direction, also.

Regards,

Reviewer

Reviewer #2: Very good article that highlights the findings on an important health issue.

Providing my suggestions to improve the quality of the paper as follows:

- writing need more correction (no spaces between many words), more clear especially in "results" section

- page 14, 1st paragraph:

- (Among all males with diabetes (N=3140)...), is a little bit confusing,

- using the word "under"

- BMI group (18.5 - 23) figure is not right in the first line) please rephrase to make it more clear

- Table 3 & forward: for lean diabetes stratification by BMI index, are the categories cumulative or discrete ones like this (<18.5, then 18.5 - <23, and 23 - < 25)? please, clarify and make it more clear

- in table 4, define the "asterisks**" meaning, the p=values , to be more understandable and clear.

- Please, properly cite the "data set" used in this survey, as instructed in the original website as in this link " https://dhsprogram.com/publications/Recommended-Citations.cfm "

6. PLOS authors have the option to publish the peer review history of their article (what does this mean?). If published, this will include your full peer review and any attached files.

**Do you want your identity to be public for this peer review?** For information about this choice, including consent withdrawal, please see our Privacy Policy.

Reviewer #1: **Yes: **Chiranjivi Adhikari

Reviewer #2: **Yes: **Elhami A Ahmed

---

## [Decision Letter · Decision Letter 1]

10 Apr 2024

Socioeconomic Gradient of Lean Diabetes in India: Evidence from National Family Health Survey, 2019-21

PGPH-D-23-02099R1

Dear Dr. Singh,

We are pleased to inform you that your manuscript 'Socioeconomic Gradient of Lean Diabetes in India: Evidence from National Family Health Survey, 2019-21' has been provisionally accepted for publication in PLOS Global Public Health.

Best regards,

Zulkarnain Jaafar

Academic Editor

Reviewer Comments (if any, and for reference):

Reviewer's Responses to Questions

**Comments to the Author**

1. If the authors have adequately addressed your comments raised in a previous round of review and you feel that this manuscript is now acceptable for publication, you may indicate that here to bypass the “Comments to the Author” section, enter your conflict of interest statement in the “Confidential to Editor” section, and submit your "Accept" recommendation.

Reviewer #1: All comments have been addressed

Reviewer #2: All comments have been addressed

2. Does this manuscript meet PLOS Global Public Health’s publication criteria? Is the manuscript technically sound, and do the data support the conclusions? The manuscript must describe methodologically and ethically rigorous research with conclusions that are appropriately drawn based on the data presented.

Reviewer #1: Yes

Reviewer #2: Yes

3. Has the statistical analysis been performed appropriately and rigorously?

Reviewer #1: Yes

Reviewer #2: Yes

4. Have the authors made all data underlying the findings in their manuscript fully available (please refer to the Data Availability Statement at the start of the manuscript PDF file)?

Reviewer #1: Yes

Reviewer #2: Yes

5. Is the manuscript presented in an intelligible fashion and written in standard English?

Reviewer #1: Yes

Reviewer #2: Yes

6. Review Comments to the Author

Reviewer #1: Thank you for the update. I am recommending to the Editor.

Reviewer..

Reviewer #2: Dear authors

Thanks a lot for addressing our suggestions professionally and comprehensively.

The manuscript became more clear

7. PLOS authors have the option to publish the peer review history of their article (what does this mean?). If published, this will include your full peer review and any attached files.

**Do you want your identity to be public for this peer review?** For information about this choice, including consent withdrawal, please see our Privacy Policy.

Reviewer #1: **Yes: **Chiranjivi Adhikari

Reviewer #2: **Yes: **Elhami A Ahmed
